# Limits and Gains of Test-Time Scaling in Vision-Language Reasoning

## Abstract

Test-time scaling (TTS) has emerged as a powerful paradigm for improving the reasoning ability of Large Language Models (LLMs) by allocating additional computation at inference, yet its application to multimodal systems such as Vision-Language Models (VLMs) remains underexplored. In this work, we present a systematic empirical study of inference-time reasoning methods applied across both open-source and closed-source VLMs on different benchmarks. Our results reveal that while closed-source models consistently benefit from structured reasoning and iterative Self-Refinement, open-source VLMs show inconsistent behavior: external verification provides the most reliable gains, whereas iterative refinement often degrades performance. We further find that the effectiveness of TTS is dataset-dependent, yielding clear improvements on multi-step reasoning tasks but offering only limited gains on perception-focused benchmarks. These findings demonstrate that TTS is not a universal solution and must be tailored to both model capabilities and task characteristics, motivating future work on adaptive TTS strategies and multimodal reward models.

## 1 Introduction

Recent advances in Large Language Models (LLMs) and AI disciplines such as natural language processing (NLP) and perceptual AI have given rise to Large Vision-Language Models (LVLMs). These multimodal models possess powerful capabilities for jointly interpreting visual and textual data Li et al. (2025b), enabling tasks ranging from image captioning and visual question answering to video analysis and applications in medical imaging and diagnostics. Given that many LVLMs leverage LLMs as a core component, techniques developed in the language modeling domain are typicall applicable to the development of VLMs. One such emerging paradigm is test-time scaling (TTS), which enhances model performance during inference by strategically increasing computation without modifying model parameters. Despite the impressive abilities of modern VLMs, their deployment in resource-sensitive settings remains challenging due to substantial inference costs Sharshar et al. (2025). These costs are influenced primarily by the parameter size of the integrated LLM, and the number of visual tokens required for image representation Li et al. (2025a). Additionally, with the limitations of further scaling foundation models due to data scarcity and performance drops under distribution shifts, interest in adaptive inference time strategies is growing. While TTS has proven effective in LLMs, improving performance on complex reasoning tasks such as mathematical problem-solving and code generation, its application to VLMs remains underexplored. Existing TTS strategies typically involve methods like parallel generation, self-refinement mechanisms, or controlled inference time using like budget forcing Zhang et al. (2025) or structured search algorithms. However, due to the inherently multimodal structure of VLMs, comprising visual encoders, language components, and fusion modules, TTS methods cannot be transferred directly and require adaptation or redesign to align with the multimodal inference process.

In this work, we present a systematic empirical study of inference-time reasoning methods, including Chain-of-Thought prompting Wei et al. (2023), Best-of-N sampling Snell et al. (2024); Bai et al. (2023); Song et al. (2025), Self-Consistency Wang et al. (2023), Self-Refinement Madaan et al. (2023), and Beam Search Xie et al. (2023), applied across both open-source and closed-source VLMs on the MathVista, MMMU, and MMBench

benchmarks. Our results reveal three key findings: (1) high-capability closed-source models consistently benefit from structured reasoning and iterative refinement, with Self-Refinement yielding the largest gains; (2) open-source VLMs display heterogeneous behaviors, where sampling-based approaches such as Best-of-N with external verification are effective, while iterative refinement often degrades performance due to unstable reasoning dynamics; and (3) the effectiveness of TTS strategies is strongly task-dependent, with multi-step reasoning tasks showing substantial improvements while perception-centric benchmarks exhibit only narrow gains. These findings demonstrate that TTS is not a universal solution for multimodal models: effective deployment requires matching the inference-time method to both model capacity and task structure. We conclude by outlining promising directions for adaptive TTS frameworks and multimodal reward modeling to further enhance reliability and efficiency in VLM inference.

## 2 Related Works

### 2.1 VLMs

Recent advances in multimodal artificial intelligence, particularly in Vision-Language Models, have been propelled by the growing availability of large-scale image-text datasets and innovations in neural architectures. These models aim to bridge the semantic gap between visual and linguistic modalities, enabling applications such as image captioning, visual question answering (VQA), and Zero-Shot classification.

A foundational contribution in this space is Contrastive Language-Image Pre-training (CLIP) Radford et al. (2021) which introduced a scalable contrastive learning approach for aligning image and text embeddings. CLIP uses dual encoders, a visual encoder and a textual encoder, to project image-text pairs into a shared space, optimizing for high similarity between matching pairs.

Large Language and Vision Assistant (LLaVA) Liu et al. (2023) further advances instruction tuning for multimodal models by combining CLIP visual encoders with open-source LLaMA-based LLMs via a projection layer. A key contribution is the use of GPT-4 to reformulate traditional image-text datasets (e.g., COCO captions) into conversational instruction-following formats. LLaVA is trained in two stages: alignment pre-training and fine-tuning on this new instruction-style data, resulting in a general-purpose visual assistant.

Qwen-VL Bai et al. (2023) offers a suite of flexible Vision-Language Models that support both general tasks and interactive applications. Its successor, Qwen2-VL Wang et al. (2024), improves image resolution handling and perceptual alignment via a refined Q-Former cross-attention module and integrates with the Qwen2 LLM. The latest version, Qwen2.5-VLBai et al. (2025), emphasizes visual reasoning and OCR-related capabilities, excelling at tasks involving textual information in images such as document understanding and chart analysis.

InternVL Chen et al. (2024c) introduces a progressive alignment of a large 6B-parameter ViT (InternViT) with LLMs through contrastive, generative, and supervised fine-tuning, achieving strong performance on 32 visual-linguistic tasks including zero-shot classification, image-text retrieval, and multimodal dialogue. InternVL1.5 Chen et al. (2024b) supports high-resolution images and bilingual datasets. InternVL2.5 Chen et al. (2025) further improves training via progressive scaling with smaller LLMs, random JPEG compression, loss reweighting, and Mixed Preference Optimization (MPO) on multimodal preference data to enhance reasoning and alignment.

Complementing this direction, Mulberry Yao et al. (2024) emphasizes structured reasoning and reflection in VLMs. It introduces a Collective Monte Carlo Tree Search (CoMCTS) mechanism where multiple model instances collaboratively explore reasoning trees, simulate feedback, and select promising solution paths. The Mulberry-260k dataset, consisting of reasoning-tree examples, enables models to learn verifiable, step-by-step reasoning, improving transparency and correctness over conventional chain-of-thought methods.

Finally, proprietary models like GPT-4 OpenAI et al. (2024b), GPT-4o OpenAI et al. (2024a), Gemini Team et al. (2025), and Claude-3 The represent the current frontier in multimodal AI. While architectural specifics remain undisclosed, these models demonstrate exceptional abilities in image understanding, multimodal reasoning, and interactive content generation.

## 2.2 Benchmarks

A variety of benchmarks have been introduced for evaluating Vision-Language Models. However, these evaluations typically focus on zero-shot question answering scenarios and do not explore method-level adaptations or assess techniques such as test-time scaling. In contrast, our work extends beyond simple evaluation settings to examine the impact of dynamic adaptation strategies during inference.

MMT-Bench Ying et al. (2024) is a comprehensive benchmark comprising over 31,000 visual multiple-choice questions, spanning 32 distinct meta-tasks. It is specifically designed to evaluate expert-level vision-language reasoning capabilities in LVLMs, such as GPT-4V. A key focus of MMT-Bench is to expose out-of-domain (OoD) vulnerabilities, making it an important tool for testing model generalization. MathVista Lu et al. (2024) assesses mathematical reasoning within visual contexts, featuring 6,141 examples drawn from 31 different datasets. This benchmark requires models to engage in complex tasks, such as algebra and geometry, demanding a high level of compositional reasoning and deep visual understanding to solve problems accurately. AI2D Kembhavi et al. (2016) is a more specialized dataset that centers on diagram-based visual question answering. Despite its smaller size, AI2D offers richly annotated diagrams that facilitate the evaluation of structured visual reasoning, making it particularly useful for studying model performance on educational and scientific illustrations. MMBench Liu et al. (2024) is a bilingual benchmark, covering both English and Chinese, with more than 3,000 questions across 20 skill domains. It employs a hierarchical taxonomy and introduces a novel evaluation methodology. A notable extension of this benchmark, Creation-MMBench, is designed to test models' abilities in generating creative, open-ended responses. Mega-bench Chen et al. (2024a) provides a highly diverse evaluation framework with over 8,000 samples encompassing more than 500 real-world tasks. It supports a wide array of output formats, including text, numerical answers, code, and LaTeX, thus facilitating a broad and flexible assessment of multimodal model capabilities across numerous application areas. BLINK Fu et al. (2024) focuses on foundational visual perception abilities, such as depth estimation and visual correspondence. It includes 3,807 multiple-choice questions that emphasize non-linguistic aspects of visual understanding, providing insight into how well models perceive spatial and relational visual cues without relying on language. MMMU Yue et al. (2024) is a large-scale benchmark geared toward evaluating college-level multimodal reasoning. With 11,500 questions drawn from academic materials, it spans six major disciplines, 30 subjects, and 183 subfields. The benchmark places particular emphasis on domain-specific visual reasoning, simulating the complexity and breadth of university-level syllabus. ScienceQA Lu et al. (2022) contains 21,000 multimodal questions that integrate images, text, and detailed explanations. Developed for K–12 science education, it supports interpretability and encourages models to employ Chain of Thought (CoT) reasoning Wei et al. (2022). This makes ScienceQA especially suitable for analyzing model transparency and the step-by-step logic behind their answers in scientific domains.

## 3 Test-time scaling methods in VLM

### 3.1 Chain of Thought

In this method, we prompt the model to generate a reasoning path (a chain of thoughts) before arriving at an answer, rather than responding directly. LLMs have demonstrated significant improvements on reasoning tasks and benchmarks using this approach. By simply adding a brief instruction before the question, we can encourage the model to think step by step, reducing hallucinations and improving response accuracy.

In this work, we extend CoT reasoning to LVLMs to evaluate its impact on reasoning-based tasks. Specifically, we prepend a prompt that guides the model to break the question into simpler components and extract relevant details from both the image and the text before answering. For multiple-choice questions, the model can be instructed to prevalidate choices before selecting a final answer, which can also be formatted explicitly for consistency.

### 3.2 Best-of-N

This method extends CoT reasoning by generating $N$ independent reasoning paths using the same CoT prompt. The final answer is then selected based on a reward mechanism, where the response with the

highest score is returned as the model's final output. Multiple strategies can be employed for both answer selection and verification.

For open-source Vision-Language Models, we utilize both **internal confidence** and **external verification** to evaluate responses. In contrast, for closed-source models, only external verification is feasible. In open-source models, confidence can be estimated by computing the log-likelihood of each answer using the model's output logits. For both open- and closed-source models, an external model can be used to assess the quality of each answer-question pair and assign a final score. However, using the same model that generated the answer as the verifier can lead to biased evaluations, as the model may overrate its own responses.

Once each answer is assigned a reward, the scores are aggregated. For example, if the answer "A" appears twice with rewards of 1 and 0.5, its final aggregated reward is computed as:

$$\text{Final Reward}(A) = \frac{1 + 0.5}{2} = 0.75.$$

After aggregation, the answer with the highest score is selected as the final output. Also it is possible not to aggregate the scores and just pick the answer with highest score.

### 3.3 Self-Consistency

Self-consistency, similar to Best-of-N, generates multiple candidate outputs, but differs primarily in the selection strategy. Instead of using a reward model to score each response, Self-Consistency selects the final answer through a majority vote. It generates $N$ independent reasoning paths using the same CoT prompt. However, rather than evaluating responses based on confidence scores or external verification, Self-Consistency selects the answer that appears most frequently across all reasoning paths.

This method is particularly useful in scenarios where assigning a numerical reward to each response is either computationally expensive or unreliable. For example, if reward models introduce noise or fail to consistently improve performance, majority voting provides a robust alternative by leveraging the model's own consistency. Self-Consistency is especially effective in reasoning tasks where different chains of thought may lead to the same correct conclusion. By aggregating multiple reasoning trajectories, it mitigates the influence of stochastic variations in model output, leading to more stable and accurate predictions.

### 3.4 Self-Refinement

Self-Refinement is an iterative framework that mimics the human editing process. Unlike single-pass generation methods (such as standard CoT) or parallel aggregation methods (such as Self-Consistency), Self-Refinement operates sequentially. In this paradigm, the model is tasked with evaluating its own output and improving it based on self-generated feedback.

The process begins with an initial generation derived from the input query. Subsequently, the model enters a feedback loop where it acts as a critic. The model is prompted to review the previous answer given the original question, identifying potential errors, logical gaps, or areas for improvement. If the model identifies that refinement is necessary, it generates a feedback signal (critique) and uses this to produce a revised answer.

This cycle repeats iteratively by generating feedback and refining the response until one of two stopping conditions is met:

1. The model determines that no further refinement is needed (convergence).

2. A pre-defined maximum number of iterations ($T_{max}$) is reached to prevent infinite loops.

This method effectively leverages the model's ability to critique and reasoning capabilities to correct hallucinations and improve the logical flow of the answer without requiring ground-truth labels or external reward models.

The algorithmic procedure for Self-Refinement is formally described in Algorithm 1.

---

**Algorithm 1** Self-Refinement Mechanism

---

**Input:** Input Query $x$, Large Language Model $\mathcal{M}$, Maximum Iterations $T_{max}$
**Output:** Refined Answer $y$

1: $y_0 \leftarrow \mathcal{M}(x)$                 ▷ Generate initial draft
2: $t \leftarrow 0$
3: **while** $t < T_{max}$ **do**
4:      $f_t \leftarrow \mathcal{M}(x, y_t, \text{"Feedback"})$         ▷ Generate feedback/critique
5:      **if** $f_t$ indicates *"No refinement needed"* **then**
6:          **break**              ▷ Convergence achieved
7:      **end if**
8:      $y_{t+1} \leftarrow \mathcal{M}(x, y_t, f_t, \text{"Refine"})$     ▷ Generate improved answer based on feedback
9:      $t \leftarrow t + 1$
10: **end while**
11: **return** $y_t$

---

## 3.5 Beam Search

The goal is to generate a high-quality reasoning path that leads to the correct final answer. To achieve this, we explore two primary strategies for evaluating candidate reasoning steps. One based on the model's internal confidence (applicable in open-source settings) and the other relying on external evaluation through an auxiliary verifier. This section describes each approach's design.

### 3.5.1 Confidence-Based Beam Search

In open-source implementations of VLMs, the model often provides token-level probability scores for generated text. These scores can be used to compute the quality of each reasoning step as follows:

**Candidate Generation:** At each reasoning step, the model is prompted to extend the current reasoning chain by generating multiple candidate continuations. For example, the prompt may explicitly instruct the model to "analyze the image" or "reason step by step" without prematurely answering the question. The prompt is dynamically constructed by appending new directives to the existing reasoning context, ensuring that each new candidate builds upon previously generated insights.

**Internal Scoring Mechanism:** We compute an aggregated token probability for each candidate from the model's output. This score reflects the model's confidence in the generated text. By aggregating these token-level scores, typically through summing the logarithms of individual probabilities, we obtain a cumulative score for the entire reasoning path.

**Ranking and Beam Pruning:** The cumulative scores serve as an internal metric to rank the candidate's reasoning paths. Beam search retains only the top candidates (as defined by a predetermined beam width) to continue to subsequent reasoning steps. This mechanism efficiently narrows the search space while leveraging the model's evaluation of its outputs.

The confidence-based approach is computationally efficient and straightforward, capitalizing on the inherent probabilistic outputs of open-source VLMs. However, its reliance solely on the model's internal signals may not capture nuanced logical consistency or coherence errors, particularly in complex reasoning tasks.

### 3.5.2 Verifier-Based Beam Search

An external verifier is employed when the internal token scores are inaccessible or a more robust evaluation is required. This verifier is instantiated as an independent model that evaluates the quality of candidate reasoning steps. We propose a verifier-based strategy:

**Candidate Generation:** Similar to the confidence-based approach, multiple candidate reasoning steps are generated at each step using the primary VLM. The candidate prompt is carefully designed to include

explicit instructions such as "Analyze the Image" or "Provide the Final Answer." The current reasoning context is preserved and extended in each prompt with a directive tailored to the specific reasoning stage.

**External Scoring Process:** Each candidate is evaluated by an auxiliary model, which assigns a qualitative score. This score is determined by mapping linguistic evaluations such as "excellent," "good," "fair," "poor," or "bad" to numerical values (e.g., 1.0, 0.75, 0.5, 0.25, 0.0, respectively). The qualitative nature of this assessment allows the verifier to focus on aspects such as clarity, logical flow, and overall reasoning quality.

**Ranking and Beam Selection:** The external scores are aggregated over the reasoning steps (using logarithmic accumulation) to form a cumulative log probability for each candidate path. Beam search then selects the top-ranked candidates to advance to the next reasoning step. This method provides an independent quality check that complements the primary model's outputs.

# 4 Experiments and Results

This section details the experimental setup, methodologies, and the results obtained from evaluating various Vision-Language Models augmented with different test-time reasoning methods across several challenging multimodal benchmarks: MathVista, MMMU, and MMBench. Our objective is to comprehensively analyze the impact and efficacy of techniques on VLM performance, identifying trends and model-specific behaviors.

## 4.1 Experimental Setup

Our experiments were conducted on a diverse set of state-of-the-art VLMs, including commercial models like Gemini 2.0 flash, GPT-4o mini, and Claude-3-Haiku, alongside open LLMs alternatives such as InternVL2.5-8B, Mulberry-8b, and Qwen2.5-VL-7B-Instruct. For each model, we evaluated its performance under a baseline (zero-shot) setting and then applied various test-time methods. The specific methods applied varied slightly across models due to practical limitations (e.g., We cannot get internal confidence for closed LLMs). The evaluation metrics for all datasets are accuracy scores, representing the percentage of correctly answered questions.

## 4.2 Comparison on MathVista Dataset

The MathVista dataset primarily assesses mathematical reasoning and visual understanding.

Table 1: Performance (Accuracy %) on MathVista Dataset

| Method | Open-Source Models | | | Closed-Source Models | |
|---|---|---|---|---|---|
| | QwenVL | Mulberry | InternVL | Gemini | GPT |
| Zero-Shot | 68.25 | 59.72 | 63.03 | 80.09 | 64.45 |
| CoT | 69.67 | 62.09 | 63.03 | 84.83 | 65.87 |
| Best-of-N | 75.36 | 64.45 | 66.35 | 85.78 | 72.51 |
| Self-Consistency | **79.62** | **70.62** | **73.46** | 84.36 | 68.25 |
| Beam Search | 68.72 | 59.72 | 62.09 | 81.52 | 64.45 |
| Self-Refinement | 67.77 | 57.82 | 60.19 | **89.57** | **72.99** |

In Table 1, closed frontier LVLMs show greater gains from TTS than open LVLMs. On closed LVLMs, TTS methods (except to Beam Search) generally lead to noticeable improvements (relative to CoT), but the magnitude of these gains vary substantially. Both of these models benefit from structured reasoning approaches such as Self-Refinement, whereas open LVLMs not only fail to gain from this TTS method but even show decreases in performance. These results suggest that stronger base models are better able to critique and iteratively improve their own outputs. Moreover, GPT compared to Gemini gains higher from Best-of-N as a sampling-based strategy, indicating that its internal reasoning paths are diverse and can be effectively exploited through selection mechanisms.

In contrast, open LLMs show more heterogeneous behavior. While some sampling-based methods like Best-of-N reliably help these models, other strategies, especially Self-Refinement, which relies solely on the model's own outputs for self-correction, can be counterproductive. This suggests that weaker or less aligned base models may not yet possess sufficiently reliable internal reasoning processes for self-correction loops to be beneficial. Importantly, all results for open models were obtained without using any closed-source verifiers; the performance differences reflect the models' intrinsic behavior. Interestingly, Self-Consistency tends to benefit open LVLMs more than closed ones, perhaps because variance among their reasoning paths can cancel out individual errors when aggregated.

Taken together, the results highlight a clear trend: frontier models leverage more complex inference-time reasoning strategies effectively and robustly, while open LVLMs benefit most from simpler sampling-driven techniques and struggle with reflective or multi-stage refinement methods.

### 4.3 MMMU Dataset

The MMMU dataset evaluates multimodal multi-discipline understanding, covering a broad range of knowledge and reasoning. Our findings are presented in Table 2.

Table 2: Performance (Accuracy %) on MMMU Dataset

| Method | Open-Source Models | | | Closed-Source Models | |
|---|---|---|---|---|---|
| | QwenVL | Mulberry | InternVL | Gemini | Claude |
| Zero-Shot | 36.7 | 28.6 | 39.5 | 61.0 | 32.4 |
| CoT | 38.1 | 39.0 | **45.2** | 61.4 | 36.7 |
| Best-of-N (confidence) | 42.4 | 35.7 | 41.9 | – | – |
| Best-of-N (verifier) | **47.1** | 37.1 | 43.3 | **67.6** | **43.8** |
| Beam Search | 39.1 | 34.0 | 41.7 | 62.0 | 38.6 |
| Self-Refinement | 37.6 | 38.1 | 40.0 | **67.6** | 42.9 |
| Self-Consistency | 45.2 | **40.5** | 42.9 | 62.5 | 38.6 |

The Best-of-N (confidence-based) results are omitted for closed-source models (Gemini and Claude) because these models do not expose token-level probabilities or calibrated confidence scores required to rank sampled candidate answers. Such internal scoring signals are accessible only in open-source models, where full logits or confidence distributions can be extracted.

For Beam Search, we report confidence-based results only for open-source models. In closed-source settings, beam search must rely on external verification because the models do not provide per-step likelihoods, making confidence-guided beam expansion infeasible. Instead, we employ a verifier-based selection for closed-source models, where candidate solutions are generated independently and ranked using an external verifier model.

Similar to the MathVista dataset, nearly all models benefit substantially from CoT prompting, highlighting its effectiveness in enhancing model performance. Overall, closed models tend to benefit more from TTS-based methods (relative to CoT) than open models. Self-Consistency yields greater improvements than Best-of-N (Confidence Based) decoding for open models, reinforcing the observation that the model's internal confidence score is not a reliable indicator of correctness. This hypothesis is further supported by the Beam Search results for open models: Beam Search performs poorly across all three evaluated models, again emphasizing the inefficiency of relying on internal confidence estimates. Additionally, Best-of-N (Verifier Based) consistently outperforms Best-of-N (Confidence Based) across all models when using Gemini 2 as the external verifier, further confirming that external verification is more effective than relying solely on the model's confidence.

Moreover, Self-Refinement provides significant gains for closed models, mirroring its impact on MathVista, and further validates the claims established in that section. However, in contrast to MathVista, CoT delivers particularly strong performance improvements on MMMU, suggesting that MMMU tasks may align more closely with the strengths of step-by-step reasoning or that the dataset inherently favors multi-step deductive

processes. This contrast also indicates that different benchmark characteristics can modulate the effectiveness of inference-time techniques.

Overall, these findings suggest that while inference-time reasoning methods such as CoT, Self-Consistency, and Self-Refinement consistently improve performance, their relative effectiveness varies across tasks and model families. This underlines the importance of dataset-specific analysis and the need for adaptive inference-time strategies tailored to both model architecture and task structure.

### 4.4 MMBench Dataset (Categorical Analysis)

The MMBench dataset provides a fine-grained evaluation across numerous categories, allowing us to inspect the performance of test-time methods on specific VLM capabilities. The results for Qwen2.5-VL and Gemini 2 Flash are presented as fractions indicating correct answers out of the total questions in each category.

For the MMBench dataset, we focus on evaluating Qwen2.5-VL and Gemini 2 Flash, which were identified in our experiments with previous datasets as the strongest performers among both closed- and open-source models. Unlike prior sections where multiple models were compared, the goal here is not to benchmark a wide range of models, but rather to analyze how these top-performing models respond to different categories of tasks. By limiting the comparison to the best models, we can more clearly observe the effects of test-time methods across diverse VLM capabilities without confounding results from weaker models.

Table 3: Combined MMBench-QwenVL Results: Comprehensive Reasoning Breakdown

| Category | Metric | Zero-Shot | CoT | Best-of-N | Self-Cons. |
|---|---|---|---|---|---|
| Fine-Grained Perception | Action Recognition | **8/9** | 7/9 | **8/9** | **8/9** |
| | Attribute Comparison | 8/9 | 8/9 | **9/9** | 8/9 |
| | Spatial Relationship | **3/4** | **3/4** | 2/4 | **3/4** |
| | Social Relation | **9/9** | **9/9** | **9/9** | **9/9** |
| | Physical Relation | 6/9 | **7/9** | 5/9 | **7/9** |
| | Physical Property | **6/9** | 5/9 | 5/9 | 6/9 |
| OCR & Coarse Perception | OCR | 3/7 | **4/7** | 3/7 | **4/7** |
| | Object Localization | 7/9 | 4/9 | **8/9** | **8/9** |
| | Nature Relation | 8/9 | 7/9 | 6/9 | **9/9** |
| | Image Topic | **9/9** | **9/9** | **9/9** | **9/9** |
| | Image Style | 8/9 | 8/9 | 8/9 | **9/9** |
| | Image Scene | **9/9** | 8/9 | **9/9** | 8/9 |
| | Image Quality | 2/9 | **4/9** | **4/9** | **4/9** |
| | Image Emotion | **8/9** | **8/9** | **8/9** | **8/9** |
| | Identity Reasoning | 7/9 | **8/9** | **8/9** | 7/9 |
| High-Level Reasoning | Future Prediction | 8/9 | 5/9 | 5/9 | **9/9** |
| | Function Reasoning | **2/2** | **2/2** | **2/2** | **2/2** |
| | Celebrity Recognition | 7/9 | **8/9** | 7/9 | **8/9** |
| | Attribute Recognition | **9/9** | **9/9** | **9/9** | **9/9** |
| | Struct. Image-Text | **5/5** | **5/5** | **5/5** | **5/5** |

The MMBench results provide a granular view of how test-time methods affect performance across diverse categories for Qwen2.5-VL and Gemini 2 Flash. It is important to note that performance is reported as 'correct/total', indicating the number of correctly answered questions out of the total for each category.

Across all three parts of MMBench-QwenVL, we observe that classical reasoning-oriented prompting methods (e.g., Chain-of-Thought, Self-Consistency, and Best-of-N sampling) produce minimal or no improvement

Table 4: Combined MMBench-Gemini Results: Comprehensive Reasoning Breakdown

| Category | Metric | Zero-Shot | CoT | Best-of-N (Ver.) | Self-Cons. |
|---|---|---|---|---|---|
| Fine-Grained Perception | Action Recognition | **9/9** | **9/9** | 8/9 | 8/9 |
| | Attribute Comparison | 3/9 | **9/9** | **9/9** | **9/9** |
| | Spatial Relationship | 2/4 | **4/4** | **4/4** | **4/4** |
| | Social Relation | **9/9** | **9/9** | **9/9** | **9/9** |
| | Physical Relation | 6/9 | 7/9 | **8/9** | 7/9 |
| | Physical Property | **6/9** | **6/9** | 5/9 | **6/9** |
| OCR & Coarse Perception | OCR | 4/7 | **5/7** | **5/7** | **5/7** |
| | Object Localization | 7/9 | 7/9 | 7/9 | 8/9 |
| | Nature Relation | **9/9** | **9/9** | **9/9** | **9/9** |
| | Image Topic | **9/9** | **9/9** | **9/9** | **9/9** |
| | Image Style | 8/9 | 8/9 | 8/9 | 9/9 |
| | Image Scene | **9/9** | **9/9** | **9/9** | **9/9** |
| | Image Quality | **3/9** | **3/9** | **3/9** | **3/9** |
| | Image Emotion | **7/9** | **7/9** | **7/9** | **7/9** |
| | Identity Reasoning | **8/9** | **8/9** | **8/9** | **8/9** |
| High-Level Reasoning | Future Prediction | 6/9 | **6/9** | **6/9** | 5/9 |
| | Function Reasoning | **2/2** | **2/2** | **2/2** | **2/2** |
| | Celebrity Recognition | **8/9** | 7/9 | 7/9 | **8/9** |
| | Attribute Recognition | 8/9 | **9/9** | **9/9** | **9/9** |
| | Struct. Image-Text | **5/5** | **5/5** | **5/5** | **5/5** |

over the strong Zero-Shot baseline. This aligns with several known characteristics of MMBench: (1) the benchmark is dominated by recognition and fine-grained perception tasks rather than multi-step reasoning; (2) most questions are closed-form, shallow, multiple-choice, where explicit reasoning does not offer additional signal; (3) stochastic decoding techniques provide little benefit on deterministic visual tasks; and (4) strong modern VLMs already saturate many of MMBench's subcategories. This suggests that for benchmarks where models like QwenVL achieve high baseline saturation, adding complex reasoning prompts often fails to provide gains and can introduce unnecessary computational overhead.

Despite this overall plateau, we do observe that in a few specific categories, inference-time methods provide measurable, albeit modest, improvements. For example, using the QwenVL model, Self-Consistency improves performance in *physical_relation* ($6/9 \rightarrow 7/9$), *ocr* ($3/7 \rightarrow 4/7$), *nature_relation* ($8/9 \rightarrow 9/9$), and *image_quality* ($2/9 \rightarrow 4/9$), while also achieving a perfect score in *image_style* ($9/9$). We observe a similar pattern with the Gemini model, where these techniques provide a slight edge in these same categories. These cases suggest that while reasoning prompts do not broadly uplift perception-limited benchmarks, they can help in categories requiring mild abstraction or multi-signal integration.

Furthermore, our findings indicate that **Self-Refinement (SR) techniques can yield slightly better performance** than CoT or Self-Consistency in these specific niches. We hypothesize this is because the errors in MMBench are primarily rooted in visual perception, not abstract logic. Methods like CoT and Self-Consistency generate multiple **independent** reasoning paths; however, if the model's initial visual percept is flawed, "voting" on multiple flawed paths is ineffective. In contrast, Self-Refinement is an **iterative** process. It allows the VLM to critique its own initial judgment and effectively "look again" at the image to verify its findings. This iterative verification is better suited to correcting subtle perceptual errors (such as in *ocr* or *image_quality*) than a single-pass logical chain. Therefore, SR provides a more compatible mechanism for improving reliability on tasks that depend on careful visual inspection rather than complex, multi-step abstract reasoning.

## Conclusion

In this work, we conduct a thorough study of test-time scaling (TTS) approaches for LVLMs on a large set of reasoning and perception tasks. While TTS has been thoroughly studied in the context of LLMs, its connection to multimodal models is largely unexplored in the literature. To systematically study a broad range of reasoning methods at inference time, such as Chain-of-Thought prompting, Best-of-N sampling, Self-Consistency, Self-Refinement, and Beam Search, we utilize both open and closed Vision-Language Models.

We find that TTS methods can be reliably and consistently applied to improve the performance of high-capability, black-box Large Language Models with Self-Refinement yielding substantial improvements. We show that such models possess sufficient capabilities to critique, edit, and improve their early-stage outputs. Second, open LVLMs have heterogeneous behaviors whereby sampling methods such as Best-of-N with external verifier and Self-Consistency improve performance, while iterative reflection methods such as Self-Refinement frequently degrade performance. This shows the necessity of adapting inference-time approaches to both the model's capacity and the stability of the reasoning processes. Third, the effectiveness of any given TTS method is highly benchmark-dependent. Datasets that require multi-step reasoning (e.g. MathVista, MMMU) typically favor structured reasoning prompts. On the perception-centric benchmark MMBench, we only see small improvements in niche categories where some abstraction or iterative checking is tolerated.

Together, these findings demonstrate that TTS is not a one-size-fits-all solution for multimodal systems. Instead, the choice of inference-time strategy must consider both the underlying model family and the nature of the target task. This suggests promising directions for future work, including the development of adaptive TTS frameworks that dynamically allocate inference-time compute based on task difficulty, model confidence, or uncertainty estimates. Additionally, designing multimodal reward models tailored to visual reasoning and exploring hybrid verification schemes may further enhance the reliability and interpretability of vision-language reasoning.

Overall, our study provides empirical evidence and actionable insights for the next generation of efficient, reliable, and adaptive VLM inference pipelines.

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

# Appendix

## A Model Configurations

All models were used with their default parameters unless explicitly stated. Table 5 summarizes the temperature and maximum generation length used for each model.

| Model | Temperature | Max New Tokens |
|---|---|---|
| QwenVL | 0.9 | 1024 |
| InternVL | 0.7 | 1024 |
| Mulberry | 0.9 | 1024 |
| Closed-Source Models (Main + Verifier) | 0.8 | Default |

Table 5: Generation parameters for each model. All remaining parameters follow the default configuration of each model.

For experiments involving multiple generated samples, we used $n = 5$ for both Best-of-N and Self-Consistency, where n denotes the number of independently sampled candidate answers. We chose $n = 5$ as it provides a practical trade-off between computational cost and answer diversity.

# B  Chain of Thought Prompt

The following prompt is designed to guide the model through a structured process to ensure clarity, correctness, and logical coherence in its final answer. The model is first instructed to examine the provided image and describe the visible elements. This step ensures that the answer is grounded in the actual visual content before any reasoning begins.

Next, the model must carefully read the question and identify what specific information is required. This prevents misinterpretation and aligns the reasoning with the question's intent. The prompt explicitly asks the model to combine the visual observations and the question context to form a clear, logical chain of thought. Breaking the reasoning into steps encourages transparency and reduces errors.

The model is instructed to compare each option against its reasoning. This step enforces deliberate evaluation rather than guessing. The model must present the final answer in a fixed format. This ensures consistent output formatting for easier parsing or automatic grading.

---

**Chain-of-Thought Prompt**

You are tasked with answering a question about an image. Follow these steps carefully:

1. **Analyze the Image**: Describe what you see in the image.

2. **Understand the Question**: Carefully read and interpret the question. Identify what specific information or relationship the question is asking about.

3. **Reason Step by Step**: Combine the information from the image and the question to reason through the problem. Break down your thought process into clear steps.

4. **Evaluate the Choices**: If multiple-choice options are provided, analyze each one and determine which best matches your reasoning.

5. **Provide the Final Answer**: Choose the most appropriate answer and format it as:
   `Final answer:  (final_choice)`

---

# C  Self-Refinement Prompts

The Self-Refinement process begins with the model generating an initial answer. The model is then asked to critique its own output using a feedback prompt designed to elicit both an analysis and a binary decision on whether refinement is required. If refinement is needed, the model is prompted again to produce an improved answer based on its own feedback. This cycle continues until either no further refinement is requested or a maximum of three refinement iterations is reached.

The prompts used in this procedure were as follows.

---

**Feedback Prompt**

```
Analyze the output of the task: "{prompt}"
---
Current Output:
{current_output}
---
Provide feedback using this format:
[FEEDBACK] <your analysis>
[REFINEMENT_NEEDED] <yes/no>
```

---

---

**Refinement Prompt**

```
Improve the output based on feedback for this task: "{prompt}"
---
Original Output:
{current_output}
Feedback:
{feedback_response}
---
Provide a refined version that addresses all feedback points.
```

---

## D  External Verification

Gemini 2 Flash was used as the external verifier in all tasks requiring explicit evaluation. The verifier scored model answers using the following prompt:

---

**Verification Prompt**

```
for this question we have this answer:
{response}
Evaluate this answer on a scale of 0.25, 0.5, 0.75, or 1,
and respond with only a single number without any explanation or detail:
[Your Answer Here]
```

---

## E  Confidence Scores

Confidence scores were computed using the average log-likelihood of the generated tokens. Given logits $\ell_t(w)$ at decoding step $t$, the confidence of output $y = (y_1, \ldots, y_T)$ is:

$$\text{Confidence}(y) = \frac{1}{T} \sum_{t=1}^{T} \log p(y_t) = \frac{1}{T} \sum_{t=1}^{T} \log \left( \frac{e^{\ell_t(y_t)}}{\sum_w e^{\ell_t(w)}} \right).$$

## F  Beam Search Configuration

Beam search was applied uniformly across models. We used a beam width of 2, meaning that at each decoding step the algorithm kept the two most promising partial sequences. The maximum number of decoding steps was set to 5, which constrained the search depth and ensured computational efficiency while still allowing for meaningful exploration of alternative reasoning paths.

For all Beam Search-based reasoning experiments, we structured the model's reasoning using the following stepwise prompts, which guided the model through a disciplined sequence of analysis:

---

**Beam Search Prompt**

- **Step 1: Analyze the Image**

  ```
  1. **Analyze the Image**: Describe what you see in the image.
  - Focus only on observable elements
  - Do NOT interpret or answer the question yet
  ```

- **Step 2: Understand the Question**

  ```
  2. **Understand the Question**: Carefully read and interpret the question.
  Identify what specific information or relationship the question is
  asking about.
  - Do NOT answer the question yet
  ```

- **Step 3: Reason Step by Step**

  ```
  3. **Reason Step by Step**: Combine the information from the image and the
  question to reason through the problem. Break down your thought process
  into clear steps.
  - Do NOT jump to conclusions yet
  ```

- **Step 4: Evaluate the Choices**

  ```
  4. **Evaluate the Choices**: If multiple-choice options are provided,
  analyze each one and determine which best matches your reasoning.
  ```

- **Final Step: Provide the Final Answer**

  ```
  5. **Provide the Final Answer**: Choose the most appropriate answer based
  on your reasoning. Format your final response as follows:
  final answer: (final_choice)
  ```

