# OpenReview forum: "Limits and Gains of Test-Time Scaling in Vision-Language Reasoning"
_TMLR — Rejected by TMLR_

### Review · Reviewer_k68W · 2026-03-15

**Summary Of Contributions:**

The paper empirically studies the effectiveness of various test-time scaling methods applied to VLMs, including CoT, beam search, self-refinement, self-consistency, and best-of-N. The paper benchmarks the results on several datasets and shows that the improvement could vary case by case and there is no universal solution. Particularly, closed-source models have better TTS behavior compared to open-source models.

**Audience:**

Yes

**Audience Explanation:**

The paper provides empirical benchmarks that could be helpful for researchers in the field of inference techniques and VLM.

**Claims And Evidence:**

Yes

**Claims Explanation:**

The paper shows benchmarking results on multiple TTS methods, models, and datasets. Most of the claims are clear. However, I still have the following concerns:

1. The TTS methods applied are very loosely connected to the vision nature. Almost all of the TTS methods are just direct adaptations from LLMs, and they cannot fully utilize the visual properties of the tasks. For example, a more visual-related TTS might be to first do grounding on the images to identify/crop particular regions of target information.

2. Current evaluation completely ignores the computational costs. It would be valuable to also compare various TTS on different models, subject to certain computational cost constraints.

3. The beam search is computed with very limited choices of options, which may greatly limit its performance. Or the paper should provide arguments/evidence about why the parameter settings for beam search is appropriate.

4. Given this is a benchmarking paper, I consider the models being tested are a bit out-of-date. GPT5 and Qwen3VL have both been out for more than half a year.

**Requested Changes:**

Please address the above comments.

---

> ### Author Response · Authors · 2026-05-18
> **Response to Reviewer k68W**
>
> Dear Reviewer k68W,
>
> Thank you for your constructive feedback, your recognition of the value of our work, and your insightful suggestions for improving the paper. We are actively working to address the concerns you raised. Below we provide clarifications and interim updates on the additional experiments currently in progress.
>
> 1- Vision-Centric Test-Time Scaling Methods
>
> We agree with your observation that most evaluated TTS methods are adapted from language-model inference strategies. To address this concern, we are currently extending our benchmark with a more vision-aware TTS strategy that incorporates explicit visual grounding/localization mechanisms during inference.
> These experiments are computationally intensive and are still running, but we expect to share the results within approximately one week. We agree that this direction is important and will strengthen the discussion section of the paper accordingly.
>
> 2- Computational Cost Analysis
>
> To address the second concern, we measured the inference time of each TTS method on a single 24GB RTX 3090 GPU. We will include this analysis in the revised manuscript to provide a cost-constrained perspective:
>
> Zero-shot (Base Prompting): Takes about 0.5 seconds per sample on average.
>
> Chain of Thought (CoT): Takes about 22 seconds per sample. This significant increase is due to the generation of a much larger number of output tokens.
>
> Best-of-N (N=5) and Self-Consistency (N=5): Take approximately 2 minutes per sample (roughly 5 times the cost of CoT due to the multiple sampling passes).
>
> Self-Refinement: Takes about 1 minute per sample on average. In our setup, we cap the refinement process at a maximum of 3 iterations. Because a powerful base model often arrives at the correct answer in the first iteration, it can exit the loop early. Furthermore, the "review" step generates fewer tokens and is faster than answering the question from scratch. Therefore, Self-Refinement generally takes about half the computational time of Best-of-N or Self-Consistency, though this is heavily dependent on the base model's inherent capabilities.
>
> 3- Justification for Beam Search Parameters
>
> Regarding the beam search parameters, we actually conducted extensive preliminary experiments to find the optimal settings. Despite numerous trials to adjust the beam width, length penalties, and prompt designs, we could not find a configuration that consistently improved upon the base results.
>
> Furthermore, to ensure a fair comparison with methods like Best-of-N, we deliberately constrained the beam search parameters so that its total computational cost and time remained comparable to the other methods. Based on these exhaustive trials, we concluded that beam search is generally not a reliable or highly effective TTS method under constrained computational budgets for these specific VLM tasks.
>
> 4- Testing on Recent State-of-the-Art Models
>
> We acknowledge the rapid pace of VLM releases. To ensure our findings generalize to the most recent state-of-the-art and are not out-of-date, we are currently running our pipeline on the newly released Qwen 3.5 9B. Importantly, we are applying the new vision-centric TTS method (mentioned in point 1) to this model as well, which directly addresses both your concern regarding model novelty and visual integration simultaneously. (Qwen 3-VL was not yet available when we initially designed and began our experiments)
>
> Next Steps:
> The new experiments (vision-centric TTS and Qwen 3.5 benchmarking) are currently running. We expect to post the complete results by next week. We will notify you here as soon as the final results are posted. Thank you again for your valuable time and feedback.

---

> > ### Author Response · Authors · 2026-05-27
> > **Response to Reviewer k68W**
> >
> > Dear Reviewer k68W,
> >
> > We are writing to follow up on our previous response and share the final results of our newly completed experiments. As promised, we evaluated the recently released Qwen 3.5 model on the MathVista benchmark. Furthermore, to directly address your insightful suggestion regarding vision-centric Test-Time Scaling (TTS) methods, we integrated VISER, a TTS method explicitly designed for visual reasoning, introduced in *Visual Structures Helps Visual Reasoning* (arXiv:2506.22146).
> >
> > ### 1. New Benchmarking Results: Qwen 3.5 on MathVista
> >
> > The table below summarizes the performance of standard LLM-based TTS methods, the vision-centric VISER method under two image-gridding strategies, and a combined approach, $\text{VISER}_{\mathrm{SC}}$, which integrates VISER with Self-Consistency.
> >
> > | Method                                    | Accuracy (%) |
> > | :---------------------------------------- | -----------: |
> > | Zero-shot                                 |         72.7 |
> > | CoT                                       |         83.2 |
> > | BoN                                       |         77.3 |
> > | Self-Refine                               |         78.2 |
> > | VISER (Square Grid)                       |         80.5 |
> > | VISER (Horizontal Lines)                  |         83.8 |
> > | Self-Consistency                          |         84.5 |
> > | $\text{VISER}_{\mathrm{SC}}$ (VISER + SC) |     86.0 |
> >
> > ### 2. Consistency with Our Previous Findings
> >
> > The results on Qwen 3.5 strongly confirm our previous findings on older models. Most notably, Self-Refine ($78.2\%$) failed to improve upon standard CoT ($83.2\%$). Upon qualitative inspection, we observed the same error patterns in Qwen 3.5 as those detailed for the other models in our manuscript: once an initial hallucination or spatial misinterpretation occurs, the model struggles to meaningfully correct its own grounded visual reasoning.
> >
> > ### 3. The Promise of Vision-Centric TTS
> >
> > Implementing VISER allowed us to evaluate a TTS strategy that interacts directly with the visual modality, rather than relying solely on text-based reasoning procedures.
> >
> > The VISER results provide additional insight into the role of the visual modality in
> > test-time scaling. Applying VISER with horizontal line overlays achieves 83.8\%,
> > closely matching Self-Consistency while operating through visual input restructuring
> > rather than purely language-side sampling. In contrast, the square-grid
> > configuration obtains only 80.5\%, a 3.3 percentage-point decrease relative to
> > horizontal lines. This substantial gap between two configurations of the same
> > vision-centric TTS method indicates that visual test-time interventions are themselves
> > sensitive to their design choices and hyperparameters. In particular, the type and
> > granularity of the imposed visual structure may either support spatial grounding or
> > introduce unnecessary visual interference. Therefore, vision-centric TTS should not
> > be regarded as a universally beneficial plug-and-play strategy.
> >
> > Most notably, combining the stronger VISER configuration with Self-Consistency yields
> > the best Qwen3.5-9B result, reaching 86.0\%. These findings suggest that visual
> > structuring and language-side sampling are complementary rather than interchangeable.
> >
> > We will integrate these new results and the expanded discussion into the final revision of the paper. We sincerely thank you again for steering us toward this highly impactful direction.

---

### Review · Reviewer_AWkK · 2026-04-10

**Summary Of Contributions:**

This paper presents a systematic empirical study of test-time scaling (TTS) methods for vision-language models (VLMs). The authors evaluate a range of inference-time reasoning strategies, including Chain-of-Thought prompting, Best-of-N sampling, Self-Consistency, Self-Refinement, and Beam Search, across both open-source and closed-source VLMs on multiple benchmarks (MathVista, MMMU, and MMBench).

The main contributions can be summarized as follows:

1. A comprehensive comparison of multiple TTS methods across diverse VLMs and benchmarks.
2. Empirical findings showing that closed-source, high-capacity models consistently benefit from structured reasoning and iterative refinement, while open-source models exhibit more heterogeneous and sometimes degraded performance.
3. Evidence that the effectiveness of TTS is highly task-dependent, with significant gains on multi-step reasoning tasks but limited improvements on perception-heavy benchmarks.
4. Practical insights suggesting that TTS strategies must be tailored to both model capability and task characteristics.

**Strengths:**

* Well-structured and extensive empirical evaluation across models and tasks.
* Provides useful and actionable insights, including negative results.
* Clear experimental trends that are relevant to the community.

**Weaknesses:**

* Limited methodological novelty; the work is primarily empirical.
* Lack of deeper mechanistic or theoretical analysis explaining observed behaviors.
* Some experimental comparisons are not fully controlled (e.g., differences between open and closed models, reliance on external verifiers).

**Audience:**

Yes

**Audience Explanation:**

Yes, the findings are relevant to a broad portion of the TMLR audience, particularly researchers working on large language models, vision-language models, and inference-time optimization.

Test-time scaling is an increasingly important paradigm for improving model performance without retraining, and understanding its limitations and applicability in multimodal settings is valuable. The paper provides practical insights into when and how different TTS strategies should be applied, which can inform both research and deployment decisions.

Even though the work is primarily empirical, the negative results and nuanced observations (e.g., failure modes of Self-Refinement in weaker models, and the limited impact on perception tasks) contribute meaningful knowledge to the community.

**Broader Impact Concerns:**

The paper does not raise significant new ethical concerns beyond those already associated with large vision-language models and large-scale AI systems.

However, the increased use of test-time scaling may lead to higher computational costs and energy consumption, especially when methods like Best-of-N sampling or iterative refinement are applied extensively. This could have environmental implications if deployed at scale.

Additionally, improved reasoning performance in multimodal systems may increase the risk of misuse in applications involving sensitive visual data or automated decision-making. While these concerns are not unique to this work, a brief discussion acknowledging computational and deployment implications would strengthen the paper.

**Claims And Evidence:**

Yes

**Claims Explanation:**

The claims are generally supported by consistent experimental results across multiple benchmarks and models. The paper provides clear quantitative evidence demonstrating the differing effects of TTS methods on open-source versus closed-source VLMs, as well as across reasoning-heavy and perception-focused tasks.

The trends reported (e.g., the effectiveness of Self-Refinement for stronger models and its degradation on weaker ones, or the limited gains on perception tasks) are consistently reflected in the presented tables and analyses. The experimental setup is clearly described, and the evaluation metrics are appropriate.

However, while the empirical evidence is convincing at a surface level, the analysis remains largely descriptive. The paper does not provide deeper causal or theoretical explanations for the observed phenomena, and some comparisons may be influenced by confounding factors (e.g., differences in model capabilities or access to internal confidence scores). These limitations slightly weaken the strength of the conclusions but do not invalidate the empirical findings.

**Requested Changes:**

**Critical:**

1. Provide deeper analysis or hypotheses explaining why certain TTS methods (e.g., Self-Refinement) fail for open-source or lower-capacity models. This could include examining error patterns, reasoning instability, or calibration issues.
2. Improve experimental control to reduce confounding factors, especially in comparisons between open-source and closed-source models (e.g., consistent use of verifiers or more controlled evaluation settings).

**Important but not critical:**
3. Include statistical significance analysis or variance across runs to strengthen confidence in the reported trends.
4. Add analysis of computational cost vs. performance gains to better contextualize the practicality of TTS methods.
5. Provide more detailed ablation studies to isolate the contribution of individual components (e.g., verifier quality, number of samples in Best-of-N).

**Optional improvements:**
6. Expand discussion on how these findings could guide the design of adaptive or hybrid TTS strategies.
7. Include qualitative case studies illustrating success and failure modes of different methods.

---

> ### Author Response · Authors · 2026-05-18
> **Response to Reviewer AWkK**
>
> Dear Reviewer AWkK,
>
> Thank you for your detailed and thoughtful review. Your suggestions regarding deeper analysis, experimental control, and practicality are especially helpful, and we are actively incorporating them into the revised paper.
>
> 1- Analysis of Failure Modes in Lower-Capacity Models
>
> We agree that a deeper analysis of why certain TTS methods fail is important. We specifically examined the failure patterns of Self-Refinement in lower-capacity/open-source models and observed a recurring instability during the review stage.
>
> In our setup, the review prompt receives both the original question and the previously generated Chain-of-Thought (CoT) answer, and must decide whether refinement is necessary. We observed that weaker models frequently fail to follow the review instruction reliably when the generated CoT becomes long or complex. Instead of evaluating the previous answer, the model often starts solving the question again from scratch.
>
> This behavior introduces compounding inconsistencies across refinement iterations, where the model gradually diverges between multiple partially conflicting reasoning traces. In contrast, stronger closed-source models tend to solve many examples correctly in the first iteration, which reduces the need for repeated refinements and decreases the probability of entering unstable refinement loops. We will include a dedicated discussion of these observed failure modes, along with qualitative examples, in the revised manuscript.
>
> 2 and 5- Experimental Control and Confounding Factors
>
> We appreciate this concern and agree that controlling for confounding factors is important.
>
> For Self-Refinement, we intentionally used the same model for both answer generation and review/refinement stages. This design choice treats refinement as an intrinsic capability of the evaluated model rather than outsourcing verification to a stronger external system.
>
> For Best-of-N and Self-Consistency, closed-source models do not expose internal confidence scores or token probabilities. Therefore, we employed a shared external verifier (Gemini 2.0) across all models to maintain a consistent evaluation pipeline. Our goal was to isolate the quality of candidate generation while minimizing variance introduced by weaker verification mechanisms.
>
> We are currently running our pipeline on the newly released Qwen3.5. Because its performance is highly competitive, it will allow us to test if a strong open-source model behaves more like the closed-source models under self-refinement.
>
> 3- Statistical Significance and Variance Across Runs
>
> We agree that variance analysis would strengthen the empirical conclusions. However, due to the substantial computational cost of evaluating multiple TTS methods across several large benchmarks and models, running multiple full evaluations was not computationally feasible.
>
> That said, the benchmarks themselves contain a large number of examples (e.g., 1,000 samples in MathVista), which helps stabilize aggregate metrics and reduces sensitivity to stochastic variation across runs. In preliminary smaller-scale repeated experiments, we observed relatively low variance in overall trends.
>
> 4- Computational Cost Analysis & Border Impact
>
> To address this concern, we measured the inference time of each TTS method on a single 24GB RTX 3090 GPU. We will include this analysis in the revised manuscript to provide a cost-constrained perspective:
>
> Zero-shot (Base Prompting): Takes about 0.5 seconds per sample on average.
>
> Chain of Thought (CoT): Takes about 22 seconds per sample. This significant increase is due to the generation of a much larger number of output tokens.
>
> Best-of-N (N=5) and Self-Consistency (N=5): Take approximately 2 minutes per sample (roughly 5 times the cost of CoT due to the multiple sampling passes).
>
> Self-Refinement: Takes about 1 minute per sample on average. In our setup, we cap the refinement process at a maximum of 3 iterations. Because a powerful base model often arrives at the correct answer in the first iteration, it can exit the loop early. Furthermore, the "review" step generates fewer tokens and is faster than answering the question from scratch. Therefore, Self-Refinement generally takes about half the computational time of Best-of-N or Self-Consistency, though this is heavily dependent on the base model's inherent capabilities.
>
> 6- Adaptive or Hybrid TTS Strategies
>
> One promising direction is to develop learned controllers (e.g., reinforcement learning or bandit-based approaches) that dynamically select among TTS strategies depending on task characteristics, uncertainty, or intermediate reasoning quality. We will include discussion of these future directions in the paper.
>
> 7- Qualitative Case Studies
>
> In the revised manuscript, we will include representative success and failure cases to illustrate the instability patterns in Self-Refinement for weaker models.

---

> > ### Author Response · Authors · 2026-05-27
> > **Response to Reviewer AWkK**
> >
> > We appreciate the reviewer's request for a deeper mechanistic hypothesis. Based on our qualitative analysis of error patterns, we hypothesize that the failure of iterative TTS methods (like Self-Refinement) in open-source models is heavily driven by context-length attention degradation.During Self-Refinement, the context window accumulates massive token overhead (comprising the image embeddings, initial reasoning chains, critique templates, and multi-turn dialogue history). While open-source backbones technically support these context lengths, their effective reasoning capabilities degrade as the prompt grows. This hypothesis is strongly supported by recent concurrent findings from Du et al. (2025) (Context Length Alone Hurts LLM Performance Despite Perfect Retrieval), who demonstrated that the sheer length of an input string induces substantial reasoning degradation in open-source models independently of retrieval quality, whereas proprietary frontier models demonstrate much higher architectural robustness to this attention overhead.
> >
> >
> > Additional support is provided by Wang et al. (2026), who analyze long-context degradation in Qwen2.5-7B and identify a critical performance-collapse. Their theoretical analysis attributes this phenomenon to interacting bottlenecks, including attention dispersion, information-transmission degradation, and positional-encoding limitations, suggesting that the model’s effective usable context length can be substantially shorter than its advertised maximum context length. This result is particularly consistent with our observations for iterative refinement: even when the nominal context budget is not exceeded, the accumulated multimodal and textual history may push open-source models into a degraded reasoning regime. In the multimodal setting, this bottleneck is further exacerbated because the model must jointly attend to static visual features and an increasingly long, potentially unstable textual reasoning history, thereby contributing to the visual-logical instability observed in our experiments.

---

### Review · Reviewer_aYwS · 2026-05-04

**Summary Of Contributions:**

In this paper, the authors propose an empirical study of inference-time reasoning methods. The outcomes f the study unveil that while closed source models rely on internal mechanisms such as reasoning, the open source ones do not benefit for that and display some heterogeneous trends. All the analysis is conduced through the lenses of Test Time Scaling, a recent trendy approach for improving reasoning in VLMs.

**Audience:**

No

**Audience Explanation:**

Although the message is clear, there is a potential major background problem: the performance gap between pen-source and closed-source models. Although GPT(mini)/Claude(3-Haiku) are competitive with the open-source models (possibly because they are the cheapest versions, Gemini consistently outperforms all the other models in all the other baselines. It feels like one model is in a completely different phase than the others, and therefore any comparison proposed is relatively less useful. Also, besides the quantitative study itself, and besides the straightforward message that the greedy Self-Refinement strategy is not always helping, there is no other concrete message.

**Broader Impact Concerns:**

No section added, not needed.

**Claims And Evidence:**

No

**Claims Explanation:**

To be precise: not always.

The core claim is quite straightforward: while in closed source VLMs the trend related to self-refinement improves the performance, for the open-source models it is not the case but the trend is more chaotic. This is partially addressed in Table 1, but already not necessarily in Table 2. Also the reasoning breakdowns in Tables 3 and 4 are hard to follow and no clear message seems to emerge.

**Requested Changes:**

Overall this paper states a fact, that by itself is not wrong, but that overall does not constitute in the current state a relevant element. t is suggested that the authors better explain why this finding is relevant (are there works that explicitly claim or state differently from the main core message of this work? Which were the theoretical bases supporting such a statement? why they were wrong?) and, if the former point is cleared out, to enlarge the VLM park with more capable state of the art models and imprve on the narrative and coherence of the manuscript.

---

> ### Author Response · Authors · 2026-05-28
> **Response to Reviewer aYwS**
>
> We thank the reviewer for the thoughtful and encouraging feedback. We appreciate the reviewer’s recognition of the strengths of our work, particularly our systematic empirical analysis of inference-time reasoning methods across both open- and closed-source VLMs through the lens of Test-Time Scaling (TTS). We are also pleased that the reviewer found our observations on the differing behaviors of open and closed models meaningful. This feedback helps strengthen our work, and we will further improve the clarity and presentation of our contributions in the revised version.
>
> ### 1- Answer to "Are the claims made in the submission supported by accurate, convincing and clear evidence?: No":
> We respectfully disagree with the reviewer’s concern and believe our claims are supported by systematic experiments across multiple open- and closed-source VLMs, diverse test-time reasoning methods, and carefully selected benchmarks. Our main claim is not that all open-source models behave similarly, but rather that they exhibit more heterogeneous and less stable behavior compared to closed-source models, especially for iterative methods such as Self-Refinement. We believe this trend is consistently reflected across Tables 1 and 2.
>
> Additionally, our contributions include a unified framework and comprehensive evaluation setup for studying test-time scaling methods in VLMs. We appreciate the reviewer’s feedback regarding the clarity of Tables 3–4, and in the revised version we will improve their presentation and discussion to make the evidence and key takeaways clearer and easier to follow.
>
> In addition, we have expanded our experiments with a newer and stronger model, Qwen 3.5, on MathVista, and the new results further support our central observations. In particular, standard CoT achieves 83.2% accuracy, whereas Self-Refinement achieves only 78.2%, again failing to improve upon a simpler reasoning baseline. Through qualitative inspection, we observe error patterns consistent with those discussed in the manuscript: once the model introduces an initial hallucination or spatial misinterpretation, subsequent self-refinement steps often fail to correct the grounded visual reasoning error.
>
> Importantly, these additional experiments also allow us to evaluate a more vision-centric TTS strategy. Specifically, VISER with horizontal lines achieves 83.8%, which is competitive with the substantially more compute-intensive Self-Consistency method at 84.5%. Moreover, combining visual structuring with Self-Consistency, i.e., VISER+SC, obtains the highest overall accuracy of 86.0%. These results strengthen our argument that directly transferring text-centric TTS methods from LLMs to VLMs may be suboptimal, and that multimodal reasoning can benefit from TTS methods designed to interact explicitly with the visual input.
>
> For completeness, we will add the following new result table to the revised manuscript:
>  Method                                    | Accuracy (%) |
> | :---------------------------------------- | -----------: |
> | Zero-shot                                 |         72.7 |
> | CoT                                       |         83.2 |
> | BoN                                       |         77.3 |
> | Self-Refine                               |         78.2 |
> | VISER (Square Grid)                       |         80.5 |
> | VISER (Horizontal Lines)                  |         83.8 |
> | $\text{VISER}_{\mathrm{SC}}$ (VISER + SC) |     86.0 |
> | Self-Consistency                          |         84.5 |
>
> We will also revise the accompanying text to more explicitly distinguish: (i) improvements from standard text-based TTS methods, (ii) failure cases of iterative refinement, (iii) differences between open- and closed-source models, and (iv) the added value of vision-centric TTS methods such as VISER and VISER+SC

---

> > ### Author Response · Authors · 2026-05-28
> > **Response to Reviewer aYwS**
> >
> > ### 2- Answer to "Would at least some individuals in TMLR's audience be interested in knowing the findings of this paper?: No":
> > We thank the reviewer for the feedback. We would like to clarify that the contribution of our work goes beyond the single
> > observation that Self-Refinement does not always help open-source models. First, we present a comprehensive framework for applying and evaluating diverse test-time scaling methods in VLMs, extending techniques that were originally studied primarily in LLM settings. We believe this unified evaluation setup can serve as a valuable resource for future research in multimodal reasoning.
> > Second, we carefully selected and analyzed multiple benchmarks to study both reasoning-oriented and perception-oriented tasks under consistent experimental settings. Finally, despite computational and cost limitations, we conducted extensive experiments across diverse open- and closed-source VLMs, which allowed us to identify meaningful trends regarding the interaction between model capability, benchmark characteristics, and inference-time reasoning methods.
> > We also acknowledge the reviewer’s concern regarding the performance gap between some closed- and open-source models, and we will clarify this discussion and the scope of our conclusions more explicitly in the revised version.
> >
> > ### 3- Answer to "Requested Changes":
> > Thank you for your constructive feedback. In the final revision, we will address your points as follows:
> > First, we will strengthen the motivation and contextualize our findings more explicitly by surveying prior works that either implicitly assume or directly claim that test-time scaling methods transfer reliably from LLMs to VLMs.
> > Second, we will expand our model pool to include additional, more capable state-of-the-art VLMs in order to broaden the empirical scope and increase the generalizability of our conclusions, subject to the budget and computational resources available to us.
> > Third, we will substantially improve the narrative coherence of the manuscript, ensuring that each finding is clearly motivated.
> > We will also expand the discussion of related concurrent findings that help contextualize the behavior observed in our experiments.
> >
> > Based on our qualitative analysis of error patterns, we hypothesize that the failure of iterative TTS methods (like Self-Refinement) in open-source models is heavily driven by context-length attention degradation.During Self-Refinement, the context window accumulates massive token overhead (comprising the image embeddings, initial reasoning chains, critique templates, and multi-turn dialogue history). While open-source backbones technically support these context lengths, their effective reasoning capabilities degrade as the prompt grows. This hypothesis is strongly supported by recent concurrent findings from Du et al. (2025) (Context Length Alone Hurts LLM Performance Despite Perfect Retrieval), who demonstrated that the sheer length of an input string induces substantial reasoning degradation in open-source models independently of retrieval quality, whereas proprietary frontier models demonstrate much higher architectural robustness to this attention overhead.
> >
> >
> > Additional support is provided by Wang et al. (2026), who analyze long-context degradation in Qwen2.5-7B and identify a critical performance-collapse threshold at approximately 40–50% of its nominal 128K context window. Their theoretical analysis attributes this phenomenon to interacting bottlenecks, including attention dispersion, information-transmission degradation, and positional-encoding limitations, suggesting that the model’s effective usable context length can be substantially shorter than its advertised maximum context length. This result is particularly consistent with our observations for iterative refinement: even when the nominal context budget is not exceeded, the accumulated multimodal and textual history may push open-source models into a degraded reasoning regime. In the multimodal setting, this bottleneck is further exacerbated because the model must jointly attend to static visual features and an increasingly long, potentially unstable textual reasoning history, thereby contributing to the visual-logical instability observed in our experiments.
> >
> > We thank the reviewer again for the constructive feedback. We believe that the
> > additional experiments, clearer positioning of our claims, expanded related-work discussion, and improved presentation of the results will substantially strengthen the manuscript.

---

### Decision · Action_Editor_PuJw · 2026-06-29

**Recommendation:** Reject

**Audience:**

Yes

**Audience Explanation:**

Yes, the topic studied in this submission is timely and relevant, and the empirical findings would interest researchers working on VLMs and inference‑time optimization.

**Claims And Evidence:**

No

**Claims Explanation:**

This submission presents an empirical study of test‑time scaling (TTS) methods for vision‑language models (VLMs). The authors evaluate several inference‑time reasoning strategies across multiple open‑ and closed‑source VLMs and benchmarks.

The reviewers agree that the topic is timely and relevant, and that the empirical direction has potential. However, two reviewers raise substantial concerns regarding the clarity, completeness, and conceptual framing of the current manuscript. These concerns directly affect the strength of the claims made in the submission (which are partially supported). While the rebuttal is thoughtful and detailed, many of the proposed improvements (such as expanded model coverage, clearer tables, deeper mechanistic analysis, and integration of vision‑centric TTS methods) are not reflected in the submitted manuscript and remain as promises for a future revision.

Because these changes are substantial and would reshape key components of the paper (motivation or empirical validation, among others), the current version does not yet meet TMLR’s standards for clarity and completeness. I therefore recommend rejection of the present submission, with an invitation to resubmit a major revision that fully integrates the requested changes. Doing so will allow the authors to better support their claims and present a more cohesive and rigorous empirical study.

**Resubmission Of Major Revision:**

The authors may consider submitting a major revision at a later time.